# INFORMATION-DIRECTED EXPLORATION FOR DEEP REINFORCEMENT LEARNING

**Nikolay Nikolov**[*]
Imperial College London, ETH Zurich
`nikolay.nikolov14@imperial.ac.uk`

**Johannes Kirschner, Felix Berkenkamp, Andreas Krause**
ETH Zurich
`{jkirschner, befelix}@inf.ethz.ch, krausea@ethz.ch`

## ABSTRACT

Efficient exploration remains a major challenge for reinforcement learning. One reason is that the variability of the returns often depends on the current state and action, and is therefore heteroscedastic. Classical exploration strategies such as upper confidence bound algorithms and Thompson sampling fail to appropriately account for heteroscedasticity, even in the bandit setting. Motivated by recent findings that address this issue in bandits, we propose to use Information-Directed Sampling (IDS) for exploration in reinforcement learning. As our main contribution, we build on recent advances in distributional reinforcement learning and propose a novel, tractable approximation of IDS for deep Q-learning. The resulting exploration strategy explicitly accounts for both parametric uncertainty and heteroscedastic observation noise. We evaluate our method on Atari games and demonstrate a significant improvement over alternative approaches.

## 1 INTRODUCTION

In Reinforcement Learning (RL), an agent seeks to maximize the cumulative rewards obtained from interactions with an unknown environment. Given only knowledge based on previously observed trajectories, the agent faces the exploration-exploitation dilemma: Should the agent take actions that maximize rewards based on its current knowledge or instead investigate poorly understood states and actions to potentially improve future performance. Thus, in order to find the optimal policy the agent needs to use an appropriate exploration strategy.

Popular exploration strategies, such as $\epsilon$-greedy (Sutton & Barto, 1998), rely on random perturbations of the agent's policy, which leads to undirected exploration. The theoretical RL literature offers a variety of statistically-efficient methods that are based on a measure of uncertainty in the agent's model. Examples include upper confidence bound (UCB) (Auer et al., 2002) and Thompson sampling (TS) (Thompson, 1933). In recent years, these have been extended to practical exploration algorithms for large state-spaces and shown to improve performance (Osband et al., 2016a; Chen et al., 2017; O'Donoghue et al., 2018; Fortunato et al., 2018). However, these methods assume that the observation noise distribution is independent of the evaluation point, while in practice *heteroscedastic observation noise* is omnipresent in RL. This means that the noise depends on the evaluation point, rather than being identically distributed (*homoscedastic*). For instance, the return distribution typically depends on a sequence of interactions and, potentially, on hidden states or inherently heteroscedastic reward observations. Kirschner & Krause (2018) recently demonstrated that, even in the simpler bandit setting, classical approaches such as UCB and TS fail to efficiently account for heteroscedastic noise.

In this work, we propose to use Information-Directed Sampling (IDS) (Russo & Van Roy, 2014; Kirschner & Krause, 2018) for efficient exploration in RL. The IDS framework can be used to design exploration-exploitation strategies that balance the estimated instantaneous regret and the expected

---

[*]Work done during exchange at ETH Zurich.

information gain. Importantly, through the choice of an appropriate *information-gain function*, IDS is able to account for parametric uncertainty and heteroscedastic observation noise during exploration.

As our main contribution, we propose a novel, tractable RL algorithm based on the IDS principle. We combine recent advances in distributional RL (Bellemare et al., 2017; Dabney et al., 2018b) and approximate parameter uncertainty methods in order to develop both homoscedastic and heteroscedastic variants of an agent that is similar to DQN (Mnih et al., 2015), but uses information-directed exploration. Our evaluation on Atari 2600 games shows the importance of accounting for heteroscedastic noise and indicates that at our approach can substantially outperform alternative state-of-the-art algorithms that focus on modeling either only epistemic or only aleatoric uncertainty. To the best of our knowledge, we are the first to develop a tractable IDS algorithm for RL in large state spaces.

## 2 RELATED WORK

Exploration algorithms are well understood in bandits and have inspired successful extensions to RL (Bubeck & Cesa-Bianchi, 2012; Lattimore & Szepesvári, 2018). Many strategies rely on the "optimism in the face of uncertainty" (Lai & Robbins, 1985) principle. These algorithms act greedily w.r.t. an augmented reward function that incorporates an exploration bonus. One prominent example is the upper confidence bound (UCB) algorithm (Auer et al., 2002), which uses a bonus based on confidence intervals. A related strategy is Thompson sampling (TS) (Thompson, 1933), which samples actions according to their posterior probability of being optimal in a Bayesian model. This approach often provides better empirical results than optimistic strategies (Chapelle & Li, 2011).

In order to extend TS to RL, one needs to maintain a distribution over Markov Decision Processes (MDPs), which is difficult in general. Similar to TS, Osband et al. (2016b) propose randomized linear value functions to maintain a Bayesian posterior distribution over value functions. Bootstrapped DQN (Osband et al., 2016a) extends this idea to deep neural networks by using an ensemble of Q-functions. To explore, Bootstrapped DQN randomly samples a Q-function from the ensemble and acts greedily w.r.t. the sample. Fortunato et al. (2018) and Plappert et al. (2018) investigate a similar idea and propose to adaptively perturb the parameter-space, which can also be thought of as tracking an approximate parameter posterior. O'Donoghue et al. (2018) propose TS in combination with an uncertainty Bellman equation, which propagates agent's uncertainty in the Q-values over multiple time steps. Additionally, Chen et al. (2017) propose to use the Q-ensemble of Bootstrapped DQN to obtain approximate confidence intervals for a UCB policy. There are also multiple other ways to approximate parametric posterior in neural networks, including Neural Bayesian Linear Regression (Snoek et al., 2015; Azizzadenesheli et al., 2018), Variational Inference (Blundell et al., 2015), Monte Carlo methods (Neal, 1995; Mandt et al., 2016; Welling & Teh, 2011), and Bayesian Dropout (Gal & Ghahramani, 2016). For an empirical comparison of these, we refer the reader to Riquelme et al. (2018).

A shortcoming of all approaches mentioned above is that, while they consider parametric uncertainty, they do not account for heteroscedastic noise during exploration. In contrast, distributional RL algorithms, such as Categorical DQN (C51) (Bellemare et al., 2017) and Quantile Regression DQN (QR-DQN) (Dabney et al., 2018b), approximate the distribution over the Q-values directly. However, both methods do not take advantage of the return distribution for exploration and use $\epsilon$-greedy exploration. Implicit Quantile Networks (IQN) (Dabney et al., 2018a) instead use a risk-sensitive policy based on a return distribution learned via quantile regression and outperform both C51 and QR-DQN on Atari-57. Similarly, Moerland et al. (2018) and Dilokthanakul & Shanahan (2018) act optimistically w.r.t. the return distribution in deterministic MDPs. However, these approaches to not consider parametric uncertainty.

Return and parametric uncertainty have previously been combined for exploration by Tang & Agrawal (2018) and Moerland et al. (2017). Both methods account for parametric uncertainty by sampling parameters that define a distribution over Q-values. The former then act greedily with respect to the expectation of this distribution, while the latter additionally samples a return for each action and then acts greedily with respect to it. However, like Thompson sampling, these approaches do not appropriately exploit the heteroscedastic nature of the return. In particular, noisier actions are more likely to be chosen, which can slow down learning.

Our method is based on Information-Directed Sampling (IDS), which can explicitly account for parametric uncertainty and heteroscedasticity in the return distribution. IDS has been primarily studied in the bandit setting (Russo & Van Roy, 2014; Kirschner & Krause, 2018). Zanette & Sarkar (2017) extend it to finite MDPs, but their approach remains impractical for large state spaces, since it requires to find the optimal policies for a set of MDPs at the beginning of each episode.

## 3 BACKGROUND

We model the agent-environment interaction with a MDP $(\mathcal{S}, \mathcal{A}, R, P, \gamma)$, where $\mathcal{S}$ and $\mathcal{A}$ are the state and action spaces, $R(\mathbf{s}, \mathbf{a})$ is the stochastic reward function, $P(\mathbf{s}'|\mathbf{s}, \mathbf{a})$ is the probability of transitioning from state $\mathbf{s}$ to state $\mathbf{s}'$ after taking action $\mathbf{a}$, and $\gamma \in [0, 1)$ is the discount factor. A policy $\pi(\cdot|\mathbf{s}) \in \mathcal{P}(\mathcal{A})$ maps a state $\mathbf{s} \in \mathcal{S}$ to a distribution over actions. For a fixed policy $\pi$, the discounted return of action $\mathbf{a}$ in state $\mathbf{s}$ is a random variable $Z^\pi(\mathbf{s}, \mathbf{a}) = \sum_{t=0}^\infty \gamma^t R(\mathbf{s}_t, \mathbf{a}_t)$, with initial state $\mathbf{s} = \mathbf{s}_0$ and action $\mathbf{a} = \mathbf{a}_0$ and transition probabilities $\mathbf{s}_t \sim P(\cdot|\mathbf{s}_{t-1}, \mathbf{a}_{t-1}), \mathbf{a}_t \sim \pi(\cdot|\mathbf{s}_t)$. The return distribution $Z$ statisfied the Bellman equation,

$$Z^\pi(\mathbf{s}, \mathbf{a}) \overset{D}{=} R(\mathbf{s}, \mathbf{a}) + \gamma Z^\pi(\mathbf{s}', \mathbf{a}'), \tag{1}$$

where $\overset{D}{=}$ denotes distributional equality. If we take the expectation of (1), the usual Bellman equation (Bellman, 1957) for the Q-function, $Q^\pi(\mathbf{s}, \mathbf{a}) = \mathbb{E}[Z^\pi(\mathbf{s}, \mathbf{a})]$, follows as

$$Q^\pi(\mathbf{s}, \mathbf{a}) = \mathbb{E}\left[R(\mathbf{s}, \mathbf{a})\right] + \gamma \mathbb{E}_{P,\pi}\left[Q^\pi(\mathbf{s}', \mathbf{a}')\right]. \tag{2}$$

The objective is to find an optimal policy $\pi^*$ that maximizes the expected total discounted return $\mathbb{E}[Z^\pi(\mathbf{s}, \mathbf{a})] = Q^\pi(\mathbf{s}, \mathbf{a})$ for all $\mathbf{s} \in \mathcal{S}, \mathbf{a} \in \mathcal{A}$.

### 3.1 UNCERTAINTY IN REINFORCEMENT LEARNING

To find such an optimal policy, the majority of RL algorithms use a point estimate of the Q-function, $Q(\mathbf{s}, \mathbf{a})$. However, such methods can be inefficient, because they can be overconfident about the performance of suboptimal actions if the optimal ones have not been evaluated before. A natural solution for more efficient exploration is to use uncertainty information. In this context, there are two source of uncertainty. Parametric (*epistemic*) uncertainty is a result of ambiguity over the class of models that explain that data seen so far, while intrinsic (*aleatoric*) uncertainty is caused by stochasticity in the environment or policy, and is captured by the distribution over returns (Moerland et al., 2017).

Osband et al. (2016a) estimate *parametric uncertainty* with a Bootstrapped DQN. They maintain an ensemble of $K$ Q-functions, $\{Q_k\}_{k=1}^K$, which is represented by a multi-headed deep neural network. To train the network, the standard bootstrap method (Efron, 1979; Hastie et al., 2001) constructs $K$ different datasets by sampling with replacement from the global data pool. Instead, Osband et al. (2016a) trains all network heads on the exact same data and diversifies the Q-ensemble via two other mechanisms. First, each head $Q_k(\mathbf{s}, \mathbf{a}; \theta)$ is trained on its own independent target head $Q_k(\mathbf{s}, \mathbf{a}; \theta^-)$, which is periodically updated (Mnih et al., 2015). Further, each head is randomly initialized, which, combined with the nonlinear parameterization and the independently targets, provides sufficient diversification.

*Intrinsic uncertainty* is captured by the return distribution $Z^\pi$. While Q-learning (Watkins, 1989) aims to estimate the expected discounted return $Q^\pi(\mathbf{s}, \mathbf{a}) = \mathbb{E}[Z^\pi(\mathbf{s}, \mathbf{a})]$, distributional RL approximates the random return $Z^\pi(\mathbf{s}, \mathbf{a})$ directly. As in standard Q-learning (Watkins, 1989), one can define a distributional Bellman optimality operator based on (1),

$$\mathcal{T}Z(\mathbf{s}, \mathbf{a}) \overset{D}{:=} R(\mathbf{s}, \mathbf{a}) + \gamma Z(\mathbf{s}', \arg\max_{\mathbf{a}' \in \mathcal{A}} \mathbb{E}[Z(\mathbf{s}', \mathbf{a}')]). \tag{3}$$

To estimate the distribution of $Z$, we use the approach of C51 (Bellemare et al., 2017) in the following. It parameterizes the return as a categorical distribution over a set of equidistant atoms in a fixed interval $[V_{\min}, V_{\max}]$. The atom probabilities are parameterized by a softmax distribution over the outputs of a parametric model. Since the parameterization $Z_\theta$ and the Bellman update $\mathcal{T}Z_\theta$ have disjoint supports, the algorithm requires an additional step $\Phi$ that projects the shifted support of $\mathcal{T}Z_\theta$ onto $[V_{\min}, V_{\max}]$. Then it minimizes the Kullback-Leibler divergence $D_{\mathrm{KL}}(\Phi\mathcal{T}Z_\theta||Z_\theta)$.

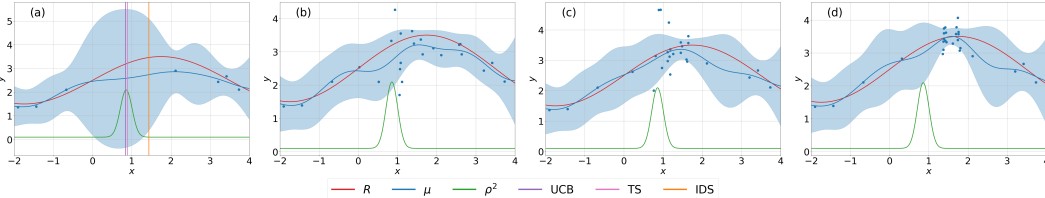

Figure 1: Gaussian Process setting. $R$: the true function, $\rho^2$: true observation noise variance, blue: confidence region with $\mu$ indicating the mean, blue dots: sampled evaluation points. (a): prior, (b), (c), (d): UCB, TS, IDS posteriors respectively after 20 samples.

## 3.2 HETEROSCEDASTICITY IN REINFORCEMENT LEARNING

In RL, heteroscedasticity means that the variance of the return distribution $Z$ depends on the state and action. This can occur in a number of ways. The variance $\mathrm{Var}(R|\mathbf{s}, \mathbf{a})$ of the reward function itself may depend on $\mathbf{s}$ or $\mathbf{a}$. Even with deterministic or homoscedastic rewards, in stochastic environments the variance of the observed return is a function of the stochasticity in the transitions over a sequence of steps. Furthermore, Partially Observable MDPs (Monahan, 1982) are also heteroscedastic due to the possibility of different states aliasing to the same observation.

Interestingly, heteroscedasticity also occurs in value-based RL *regardless of the environment*. This is due to Bellman targets being generated based on an evolving policy $\pi$. To demonstrate this, consider a standard observation model used in supervised learning $y_t = f(\mathbf{x}_t) + \epsilon_t(\mathbf{x}_t)$, with true function $f$ and Gaussian noise $\epsilon_t(\mathbf{x}_t)$. In Temporal Difference (TD) algorithms (Sutton & Barto, 1998), given a sample transition $(\mathbf{s}_t, \mathbf{a}_t, r_t, \mathbf{s}_{t+1})$, the learning target is generated as $y_t = r_t + \gamma Q^\pi(\mathbf{s}_{t+1}, \mathbf{a}')$, for some action $\mathbf{a}'$. Similarly to the observation model above, we can describe TD-targets for learning $Q^*$ being generated as $y_t = f(\mathbf{s}_t, \mathbf{a}_t) + \epsilon_t^\pi(\mathbf{s}_t, \mathbf{a}_t)$, with $f$ and $\epsilon_t^\pi$ given by

$$f(\mathbf{s}_t, \mathbf{a}_t) = Q^*(\mathbf{s}_t, \mathbf{a}_t) = \mathbb{E}[R(\mathbf{s}_t, \mathbf{a}_t)] + \gamma \mathbb{E}_{\mathbf{s}' \sim p(\mathbf{s}'|\mathbf{s}_t, \mathbf{a}_t)}[\max_{\mathbf{a}'} Q^*(\mathbf{s}', \mathbf{a}')]$$

$$\epsilon_t^\pi(\mathbf{s}_t, \mathbf{a}_t) = r_t + \gamma Q^\pi(\mathbf{s}_{t+1}, \mathbf{a}') - f(\mathbf{s}_t, \mathbf{a}_t) \tag{4}$$

$$= (r_t - \mathbb{E}[R(\mathbf{s}_t, \mathbf{a}_t)]) + \gamma \left( Q^\pi(\mathbf{s}_{t+1}, \mathbf{a}') - \mathbb{E}_{\mathbf{s}' \sim p(\mathbf{s}'|\mathbf{s}_t, \mathbf{a}_t)}[\max_{\mathbf{a}'} Q^*(\mathbf{s}', \mathbf{a}')] \right)$$

The last term clearly shows the dependence of the noise function $\epsilon_t^\pi(\mathbf{s}, \mathbf{a})$ on the policy $\pi$, used to generate the Bellman target. Note additionally that heteroscedastic targets are not limited to TD-learning methods, but also occur in TD($\lambda$) and Monte-Carlo learning (Sutton & Barto, 1998), no matter if the environment is stochastic or not.

## 3.3 INFORMATION-DIRECTED SAMPLING

Information-Directed Sampling (IDS) is a bandit algorithm, which was first introduced in the Bayesian setting by Russo & Van Roy (2014), and later adapted to the frequentist framework by Kirschner & Krause (2018). Here, we concentrate on the latter formulation in order to avoid keeping track of a posterior distribution over the environment, which itself is a difficult problem in RL. The bandit problem is equivalent to a single state MDP with stochastic reward function $R(\mathbf{a}, \mathbf{s}) = R(\mathbf{a})$ and optimal action $\mathbf{a}^* = \arg\max_{\mathbf{a} \in \mathcal{A}} \mathbb{E}[R(\mathbf{a})]$. We define the (expected) regret $\Delta(\mathbf{a}) := \mathbb{E}[R(\mathbf{a}^*) - R(\mathbf{a})]$, which is the loss in reward for choosing an suboptimal action $\mathbf{a}$. Note, however, that we cannot directly compute $\Delta(\mathbf{a})$, since it depends on $R$ and the unknown optimal action $\mathbf{a}^*$. Instead, IDS uses a conservative regret estimate $\hat{\Delta}_t(\mathbf{a}) = \max_{\mathbf{a}' \in \mathcal{A}} u_t(\mathbf{a}') - l_t(\mathbf{a})$, where $[l_t(\mathbf{a}), u_t(\mathbf{a})]$ is a confidence interval which contains the true expected reward $\mathbb{E}[R(\mathbf{a})]$ with high probability.

In addition, assume for now that we are given an *information gain function* $I_t(\mathbf{a})$. Then, at any time step $t$, the IDS policy is defined by

$$\mathbf{a}_t^{\mathrm{IDS}} \in \arg\min_{\mathbf{a} \in \mathcal{A}} \frac{\hat{\Delta}_t(\mathbf{a})^2}{I_t(\mathbf{a})}. \tag{5}$$

Technically, this is known as *deterministic* IDS which, for simplicity, we refer to as IDS throughout this work. Intuitively, IDS chooses actions with small regret-information ratio $\hat{\Psi}_t(\mathbf{a}) := \frac{\hat{\Delta}_t(\mathbf{a})^2}{I_t(\mathbf{a})}$ to balance between incurring regret and acquiring new information at each step. Kirschner & Krause (2018) introduce several information-gain functions and derive a high-probability bound on the cumulative regret, $\sum_{t=1}^{T} \Delta_t(\mathbf{a}_t^{\mathrm{IDS}}) \leq \mathcal{O}(\sqrt{T\gamma_T})$. Here, $\gamma_T$ is an upper bound on the total information gain $\sum_{t=1}^{T} I_t(\mathbf{a}_t)$, which has a sublinear dependence in $T$ for different function classes and the specific information-gain function we use in the following (Srinivas et al., 2010). The overall regret bound for IDS matches the best bound known for the widely used UCB policy for linear and kernelized reward functions.

One particular choice of the information gain function, that works well empirically and we focus on in the following, is $I_t(\mathbf{a}) = \log\left(1 + \sigma_t(\mathbf{a})^2/\rho(\mathbf{a})^2\right)$ (Kirschner & Krause, 2018). Here $\sigma_t(a)^2$ is the variance in the parametric estimate of $\mathbb{E}[R(\mathbf{a})]$ and $\rho(\mathbf{a})^2 = \mathrm{Var}[R(\mathbf{a})]$ is the variance of the observed reward. In particular, the information gain $I_t(\mathbf{a})$ is small for actions with little uncertainty in the true expected reward or with reward that is subject to high observation noise. Importantly, note that $\rho(\mathbf{a})^2$ may explicitly depend on the selected action $\mathbf{a}$, which allows the policy to account for heteroscedastic noise.

We demonstrate the advantage of such a strategy in the Gaussian Process setting (Murphy, 2012). In particular, for an arbitrary set of actions $\mathbf{a}_1, \ldots, \mathbf{a}_N$, we model the distribution of $R(\mathbf{a}_1), \ldots, R(\mathbf{a}_N)$ by a multivariate Gaussian, with covariance $\mathrm{Cov}[R(\mathbf{a}_i), R(\mathbf{a}_j)] = \kappa(\mathbf{x}_i, \mathbf{x}_j)$, where $\kappa$ is a positive definite kernel. In our toy example, the goal is to maximize $R(\mathbf{x})$ under heteroscedastic observation noise with variance $\rho(\mathbf{x})^2$ (Figure 1). As UCB and TS do not consider observation noise in the acquisition function, they may sample at points where $\rho(\mathbf{x})^2$ is large. Instead, by exploiting kernel correlation, IDS is able to shrink the uncertainty in the high-noise region with fewer samples, by selecting a nearby point with potentially higher regret but small noise.

## 4 INFORMATION-DIRECTED SAMPLING FOR REINFORCEMENT LEARNING

In this section, we use the IDS strategy from the previous section in the context of deep RL. In order to do so, we have to define a tractable notion of regret $\Delta_t$ and information gain $I_t$.

### 4.1 ESTIMATING REGRET AND INFORMATION GAIN

In the context of RL, it is natural to extend the definition of instantaneous regret of action $\mathbf{a}$ in state $\mathbf{s}$ using the Q-function

$$\Delta_t^\pi(\mathbf{s}, \mathbf{a}) := \mathbb{E}_P\left[\max_{\mathbf{a}'} Q^\pi(\mathbf{s}, \mathbf{a}') - Q_t^\pi(\mathbf{s}, \mathbf{a})|\mathcal{F}_{t-1}\right], \tag{6}$$

where $\mathcal{F}_t = \{\mathbf{s}_1, \mathbf{a}_1, r_1, \ldots \mathbf{s}_t, \mathbf{a}_t, r_t\}$ is the history of observations at time $t$. The regret definition in eq. (6) captures the loss in return when selecting action $\mathbf{a}$ in state $\mathbf{s}$ rather than the optimal action. This is similar to the notion of the advantage function. Since $\Delta_t^\pi(\mathbf{s}, \mathbf{a})$ depends on the true Q-function $Q^\pi$, which is not available in practice and can only be estimated based on finite data, the IDS framework instead uses a conservative estimate.

To do so, we must characterize the parametric uncertainty in the Q-function. Since we use neural networks as function approximators, we can obtain approximate confidence bounds using a Bootstrapped DQN (Osband et al., 2016a). In particular, given an ensemble of $K$ action-value functions, we compute the empirical mean and variance of the estimated Q-values,

$$\mu(\mathbf{s}, \mathbf{a}) = \frac{1}{K}\sum_{k=1}^{K} Q_k(\mathbf{s}, \mathbf{a}), \qquad \sigma(\mathbf{s}, \mathbf{a})^2 = \frac{1}{K}\sum_{k=1}^{K} \left(Q_k(\mathbf{s}, \mathbf{a}) - \mu(\mathbf{s}, \mathbf{a})\right)^2. \tag{7}$$

Based on the mean and variance estimate in the Q-values, we can define a surrogate for the regret using confidence intervals,

$$\hat{\Delta}_t^\pi(\mathbf{s}, \mathbf{a}) = \max_{\mathbf{a}'\in\mathcal{A}}\left(\mu_t(\mathbf{s}, \mathbf{a}') + \lambda_t\sigma_t(\mathbf{s}, \mathbf{a}')\right) - \left(\mu_t(\mathbf{s}, \mathbf{a}) - \lambda_t\sigma_t(\mathbf{s}, \mathbf{a})\right). \tag{8}$$

where $\lambda_t$ is a scaling hyperparameter. The first term corresponds to the maximum plausible value that the Q-function could take at a given state, while the right term lower-bounds the Q-value given the chosen action. As a result, eq. (8) provides a conservative estimate of the regret in eq. (6).

---

**Algorithm 1** Deterministic Information-Directed Q-learning

---

**Input**: $\lambda$, action-value function $Q$ with $K$ outputs $\{Q_k\}_{k=1}^K$, action-value distribution $Z$
**for** episode $i = 1 : M$ **do**
    Get initial state $\mathbf{s}_0$
    **for** step $t = 0 : T$ **do**
        $\mu(\mathbf{s}_t, \mathbf{a}) = \frac{1}{K} \sum_{k=1}^K Q_k(\mathbf{s}_t, \mathbf{a})$
        $\sigma(\mathbf{s}_t, \mathbf{a})^2 = \frac{1}{K} \sum_{k=1}^K [Q_k(\mathbf{s}_t, \mathbf{a}) - \mu(\mathbf{s}_t, \mathbf{a})]^2$
        $\hat{\Delta}(\mathbf{s}_t, \mathbf{a}) = \max_{\mathbf{a}' \in \mathcal{A}} [\mu(\mathbf{s}_t, \mathbf{a}') + \lambda\sigma(\mathbf{s}_t, \mathbf{a}')] - [\mu(\mathbf{s}_t, \mathbf{a}) - \lambda\sigma(\mathbf{s}_t, \mathbf{a})]$
        $\rho(\mathbf{s}_t, \mathbf{a})^2 = \mathrm{Var}\left(Z(\mathbf{s}_t, \mathbf{a})\right) / \left(\epsilon_1 + \frac{1}{|\mathcal{A}|} \sum_{\mathbf{a}' \in \mathcal{A}} \mathrm{Var}\left(Z(\mathbf{s}_t, \mathbf{a}')\right)\right)$
        $I(\mathbf{s}_t, \mathbf{a}) = \log\left(1 + \frac{\sigma(\mathbf{s}_t, \mathbf{a})^2}{\rho(\mathbf{s}_t, \mathbf{a})^2}\right) + \epsilon_2$
        Compute regret-information ratio: $\hat{\Psi}(\mathbf{s}_t, \mathbf{a}) = \frac{\hat{\Delta}(\mathbf{s}_t, \mathbf{a})^2}{I(\mathbf{s}_t, \mathbf{a})}$
        Execute action $\mathbf{a}_t = \arg\min_{\mathbf{a} \in \mathcal{A}} \hat{\Psi}(\mathbf{s}_t, \mathbf{a})$, observe $r_t$ and state $\mathbf{s}_{t+1}$
    **end for**
**end for**

---

Given the regret surrogate, the only missing component to use the IDS strategy in eq. (5) is to compute the information gain function $I_t$. In particular, we use $I_t(\mathbf{a}) = \log\left(1 + \sigma_t(\mathbf{a})^2/\rho(\mathbf{a})^2\right)$ based on the discussion in (Kirschner & Krause, 2018). In addition to the previously defined predictive parameteric variance estimates for the regret, it depends on the variance of the noise distribution, $\rho$. While in the bandit setting we track one-step rewards, in RL we focus on learning from returns from complete trajectories. Therefore, instantaneous reward observation noise variance $\rho(\mathbf{a})^2$ in the bandit setting transfers to the variance of the return distribution $\mathrm{Var}\left(Z(\mathbf{s}, \mathbf{a})\right)$ in RL. We point out that the scale of $\mathrm{Var}\left(Z(\mathbf{s}, \mathbf{a})\right)$ can substantially vary depending on the stochasticity of the policy and the environment, as well as the reward scaling. This directly affects the scale of the information gain and the degree to which the agent chooses to explore. Since the weighting between regret and information gain in the IDS ratio is implicit, for stable performance across a range of environments, we propose computing the information gain $I(\mathbf{s}, \mathbf{a}) = \log\left(1 + \frac{\sigma(\mathbf{s}, \mathbf{a})^2}{\rho(\mathbf{s}, \mathbf{a})^2}\right) + \epsilon_2$ using the normalized variance

$$\rho(\mathbf{s}, \mathbf{a})^2 = \frac{\mathrm{Var}\left(Z(\mathbf{s}, \mathbf{a})\right)}{\epsilon_1 + \frac{1}{|\mathcal{A}|} \sum_{\mathbf{a}' \in \mathcal{A}} \mathrm{Var}\left(Z(\mathbf{s}, \mathbf{a}')\right)}, \tag{9}$$

where $\epsilon_1, \epsilon_2$ are small constants that prevent division by 0. This normalization step brings the mean of all variances to 1, while keeping their values positive. Importantly, it preserves the signal needed for noise-sensitive exploration and allows the agent to account for numerical differences across environments and favor the same amount of risk. We also experimentally found this version to give better results compared to the unnormalized variance $\rho(\mathbf{s}, \mathbf{a})^2 = \mathrm{Var}\left(Z(\mathbf{s}, \mathbf{a})\right)$.

## 4.2 INFORMATION-DIRECTED REINFORCEMENT LEARNING

Using the estimates for regret and information gain, we provide the complete control algorithm in Algorithm 1. At each step, we compute the parametric uncertainty over $Q(\mathbf{s}, \mathbf{a})$ as well as the distribution over returns $Z(\mathbf{s}, \mathbf{a})$. We then follow the steps from Section 4.1 to compute the regret and the information gain of each action, and select the one that minimizes the regret-information ratio $\hat{\Psi}(\mathbf{s}, \mathbf{a})$.

To estimate parametric uncertainty, we use the exact same training procedure and architecture as Bootstrapped DQN (Osband et al., 2016a): we split the DQN architecture (Mnih et al., 2015) into $K$ bootstrap heads after the convolutional layers. Each head $Q_k(\mathbf{s}, \mathbf{a}; \theta)$ is trained against its own target head $Q_k(\mathbf{s}, \mathbf{a}; \theta^-)$ and all heads are trained on the exact same data. We use Double DQN targets (van Hasselt et al., 2016) and normalize gradients propagated by each head by $1/K$.

To estimate $Z(\mathbf{s}, \mathbf{a})$, it makes sense to share some of the weights $\theta$ from the Bootstrapped DQN. We propose to use the output of the last convolutional layer $\phi(\mathbf{s})$ as input to a separate head that estimates $Z(\mathbf{s}, \mathbf{a})$. The output of this head is the only one used for computing $\rho(\mathbf{s}, \mathbf{a})^2$ and is also *not* included in the bootstrap estimate. For instance, this head can be trained using C51 or QR-

Table 1: Mean and median of best scores computed across the Atari 2600 games from Table 3 and 4 in the appendix, measured as human-normalized percentages (Nair et al., 2015). QR-DQN and IQN scores obtained from Table 1 in Dabney et al. (2018a), by removing the scores of Defender and Surround. DQN-IDS and C51-IDS averaged over 3 seeds.

|  | Mean | Median |
|---|---|---|
| DQN | 232% | 79% |
| DDQN | 313% | 118% |
| Dueling | 379% | 151% |
| NoisyNet-DQN | 389% | 123% |
| Prior. | 444% | 124% |
| Bootstrapped DQN | 553% | 139% |
| Prior. Dueling | 608% | 172% |
| NoisyNet-Dueling | 651% | 172% |
| DQN-IDS | 757% | 187% |
| C51 | 721% | 178% |
| QR-DQN | 888% | 193% |
| IQN | 1048% | 218% |
| C51-IDS | **1058%** | **253%** |

DQN, with variance $\mathrm{Var}\,(Z(\mathbf{s}, \mathbf{a})) = \sum_i p_i(z_i - \mathbb{E}[Z(\mathbf{s}, \mathbf{a})])^2$, where $z_i$ denotes the atoms of the distribution support, $p_i$, their corresponding probabilities, and $E[Z(\mathbf{s}, \mathbf{a})] = \sum_i p_i z_i$. To isolate the effect of noise-sensitive exploration from the advantages of distributional training, we do not propagate distributional loss gradients in the convolutional layers and use the representation $\phi(\mathbf{s})$ learned only from the bootstrap branch. This is not a limitation of our approach and both (or either) bootstrap and distributional gradients can be propagated through the convolutional layers.

Importantly, our method can account for deep exploration, since both the parametric uncertainty $\sigma(\mathbf{s}, \mathbf{a})^2$ and the intrinsic uncertainty $\rho(\mathbf{s}, \mathbf{a})^2$ estimates in the information gain are extended beyond a single time step and propagate information over sequences of states. We note the difference with intrinsic motivation methods, which augment the reward function by adding an exploration bonus to the step reward (Houthooft et al., 2016; Stadie et al., 2015; Schmidhuber, 2010; Bellemare et al., 2016; Tang et al., 2017). While the bonus is sometimes based on an information-gain measure, the estimated optimal policy is often affected by the augmentation of the rewards.

## 5 Experiments

We now provide experimental results on 55 of the Atari 2600 games from the Arcade Learning Environment (ALE) (Bellemare et al., 2013), simulated via the OpenAI gym interface (Brockman et al., 2016). We exclude Defender and Surround from the standard Atari-57 selection, since they are not available in OpenAI gym. Our method builds on the standard DQN architecture and we expect it to benefit from recent improvements such as Dueling DQN (Wang et al., 2016) and prioritized replay (Schaul et al., 2016). However, in order to separately study the effect of changing the exploration strategy, we compare our method without these additions. Our code can be found at https://github.com/nikonikolov/rltf/tree/ids-drl.

We evaluate two versions of our method: a homoscedastic one, called DQN-IDS, for which we do not estimate $Z(\mathbf{s}, \mathbf{a})$ and set $\rho(\mathbf{s}, \mathbf{a})^2$ to a constant, and a heteroscedastic one, C51-IDS, for which we estimate $Z(\mathbf{s}, \mathbf{a})$ using C51 as previously described. DQN-IDS uses the exact same network architecture as Bootstrapped DQN. For C51-IDS, we add the fully-connected part of the C51 network (Bellemare et al., 2017) on top of the last convolutional layer of the DQN-IDS architecture, but we do not propagate distributional loss gradients into the convolutional layers. We use a target network to compute Bellman updates, with double DQN targets only for the bootstrap heads, but not for the distributional update. Weights are updated using the Adam optimizer (Kingma & Ba, 2015). We evaluate the performance of our method using a mean greedy policy that is computed on

the bootstrap heads

$$\arg\max_{\mathbf{a}\in\mathcal{A}} \frac{1}{K} \sum_{k=1}^{K} Q_k(\mathbf{s}, \mathbf{a}). \tag{10}$$

Due to computational limitations, we did *not* perform an extensive hyperparameter search. Our final algorithm uses $\lambda = 0.1$, $\rho(\mathbf{s}, \mathbf{a})^2 = 1.0$ (for DQN-IDS) and target update frequency of 40000 agent steps, based on a parameter search over $\lambda \in \{0.1, 1.0\}$, $\rho^2 \in \{0.5, 1.0\}$, and target update in $\{10000, 40000\}$. For C51-IDS, we put a heuristically chosen lower bound of $0.25$ on $\rho(\mathbf{s}, \mathbf{a})^2$ to prevent the agent from fixating on "noiseless" actions. This bound is introduced primarily for numerical reasons, since, even in the bandit setting, the strategy may degenerate as the noise variance of a single action goes to zero. We also ran separate experiments without this lower bound and while the per-game scores slightly differ, the overall change in mean human-normalized score was only 23%. We also use the suggested hyperparameters from C51 and Bootstrapped DQN, and set learning rate $\alpha = 0.00005$, $\epsilon_{\text{ADAM}} = 0.01/32$, number of heads $K = 10$, number of atoms $N = 51$. The rest of our training procedure is identical to that of Mnih et al. (2015), with the difference that we do not use $\epsilon$-greedy exploration. All episodes begin with up to 30 random no-ops (Mnih et al., 2015) and the horizon is capped at 108K frames (van Hasselt et al., 2016). Complete details are provided in Appendix A.

To provide comparable results with existing work we report evaluation results under the best agent protocol. Every 1M training frames, learning is frozen, the agent is evaluated for 500K frames and performance is computed as the average episode return from this latest evaluation run. Table 1 shows the mean and median human-normalized scores (van Hasselt et al., 2016) of the best agent performance after 200M training frames. Additionally, we illustrate the distributions learned by C51 and C51-IDS in Figure 3.

We first point out the results of DQN-IDS and Bootstrapped DQN. While both methods use the same architecture and similar optimization procedures, DQN-IDS outperforms Bootstrapped DQN by around 200%. This suggests that simply changing the exploration strategy from TS to IDS (along with the type of optimizer), even without accounting for heteroscedastic noise, can substantially improve performance. Furthermore, DQN-IDS slightly outperforms C51, even though C51 has the benefits of distributional learning.

We also see that C51-IDS outperforms C51 and QR-DQN and achieves slightly better results than IQN. Importantly, the fact that C51-IDS substantially outperforms DQN-IDS, highlights the significance of accounting for heteroscedastic noise. We also experimented with a QRDQN-IDS version, which uses QR-DQN instead of C51 to estimate $Z(\mathbf{s}, \mathbf{a})$ and noticed that our method can benefit from better approximation of the return distribution. While we expect the performance over IQN to be higher, we do not include QRDQN-IDS scores since we were unable to reproduce the reported QR-DQN results on some games. We also note that, unlike C51-IDS, IQN is specifically tuned for risk sensitivity. One way to get a risk-sensitive IDS policy is by tuning for $\beta$ in the additive IDS formulation $\hat{\Psi}(\mathbf{s}, \mathbf{a}) = \hat{\Delta}(\mathbf{s}, \mathbf{a})^2 - \beta I(\mathbf{s}, \mathbf{a})$, proposed by Russo & Van Roy (2014). We verified on several games that C51-IDS scores can be improved by using this additive formulation and we believe such gains can be extended to the rest of the games.

## 6   CONCLUSION

We extended the idea of frequentist Information-Directed Sampling to a practical RL exploration algorithm that can account for heteroscedastic noise. To the best of our knowledge, we are the first to propose a tractable IDS algorithm for RL in large state spaces. Our method suggests a new way to use the return distribution in combination with parametric uncertainty for efficient deep exploration and demonstrates substantial gains on Atari games. We also identified several sources of heteroscedasticity in RL and demonstrated the importance of accounting for heteroscedastic noise for efficient exploration. Additionally, our evaluation results demonstrated that similarly to the bandit setting, IDS has the potential to outperform alternative strategies such as TS in RL.

There remain promising directions for future work. Our preliminary results show that similar improvements can be observed when IDS is combined with continuous control RL methods such as the Deep Deterministic Policy Gradient (DDPG) (Lillicrap et al., 2016). Developing a computationally efficient approximation of the randomized IDS version, which minimizes the regret-information

ratio over the set of stochastic policies, is another idea to investigate. Additionally, as indicated by Russo & Van Roy (2014), IDS should be seen as a design principle rather than a specific algorithm, and thus alternative information gain functions are an important direction for future research.

ACKNOWLEDGMENTS

We thank Ian Osband and Will Dabney for providing details about the Atari evaluation protocol. This work was supported by SNSF grant 200020_159557, the Vector Institute and the Open Philanthropy Project AI Fellows Program.

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

# A  HYPERPARAMETERS

Table 2: ALE hyperparameters

| Hyperparameter | Value | Description |
|---|---|---|
| $\lambda$ | 0.1 | Scale factor for computing regret surrogate |
| $\rho^2$ | 1.0 | Observation noise variance for DQN-IDS |
| $\epsilon_1, \epsilon_2$ | 0.00001 | Information-ratio constants; prevent division by 0 |
| mini-batch size | 32 | Size of mini-batch samples for gradient descent step |
| replay buffer size | 1M | The number of most recent observations stored in the replay buffer |
| agent history length | 4 | The number of most recent frames concatenated as input to the network |
| action repeat | 4 | Repeat each action selected by the agent this many times |
| $\gamma$ | 0.99 | Discount factor |
| training frequency | 4 | The number of times an action is selected by the agent between successive gradient descent steps |
| $K$ | 10 | Number of bootstrap heads |
| $\beta_1$ | 0.9 | Adam optimizer parameter |
| $\beta_2$ | 0.99 | Adam optimizer parameter |
| $\epsilon_{\text{ADAM}}$ | 0.01/32 | Adam optimizer parameter |
| $\alpha$ | 0.00005 | learning rate |
| learning starts | 50000 | Agent step at which learning starts. Random policy beforehand |
| number of bins | 51 | Number of bins for Categorical DQN (C51) |
| $[V_{\text{MIN}}, V_{\text{MAX}}]$ | [-10, 10] | C51 distribution range |
| number of quantiles | 200 | Number of quantiles for QR-DQN |
| target network update frequency | 40000 | Number of *agent* steps between consecutive target updates |
| evaluation length | 125K | Number of *agent* steps each evaluation window lasts for. Equivalent to 500K frames |
| evaluation frequency | 250K | The number of steps the agent takes in training mode between two evaluation runs. Equivalent to 1M frames |
| eval episode length | 27K | Number of maximum *agent* steps during an evaluation episode. Equivalent to 108K frames |
| max no-ops | 30 | Maximum number no-op actions before the episode starts |

# B  SUPPLEMENTAL RESULTS

Human-normalized scores are computed as (van Hasselt et al., 2016),

$$score = \frac{agent - random}{human - random} \times 100 \qquad (11)$$

where $agent$, $human$ and $random$ represent the per-game raw scores.

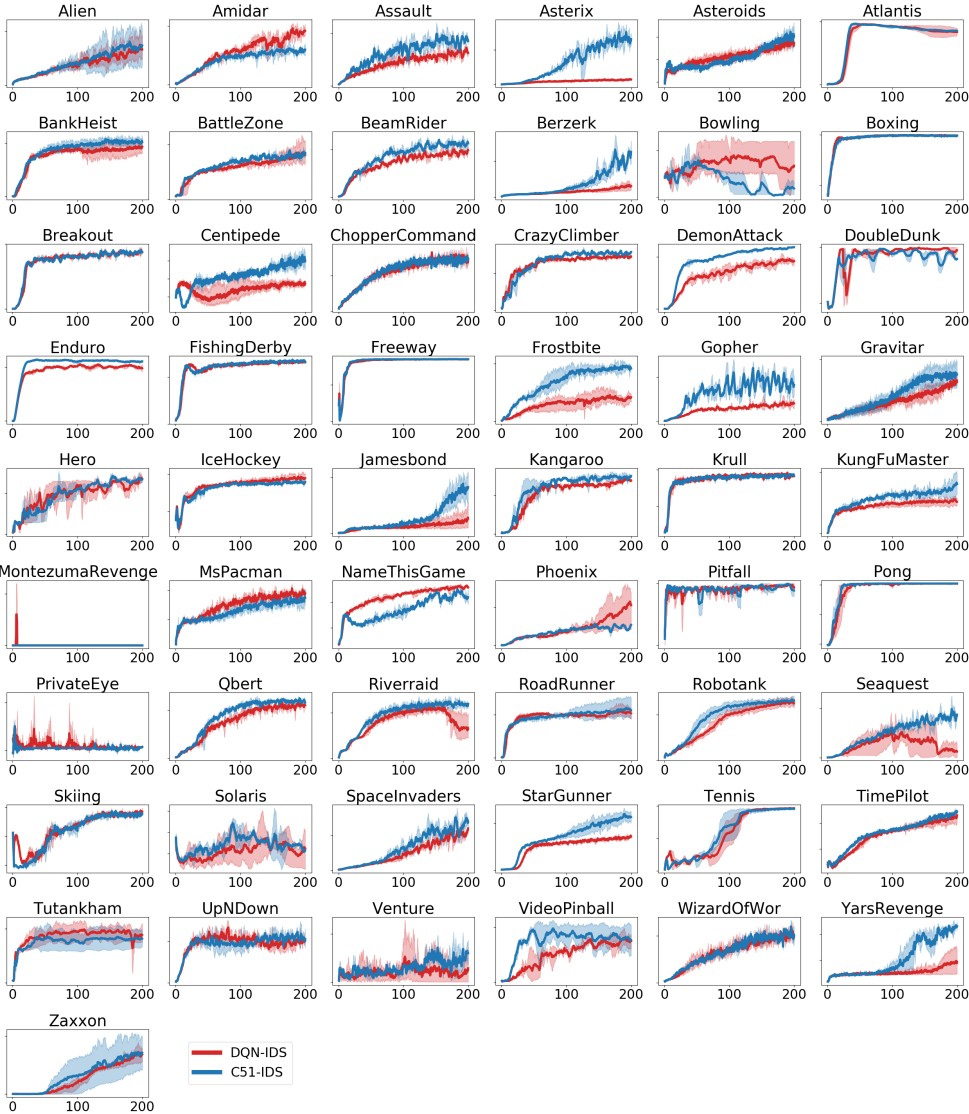

Figure 2: Training curves for DQN-IDS and C51-IDS averaged over 3 seeds. Shaded areas correspond to min and max returns.

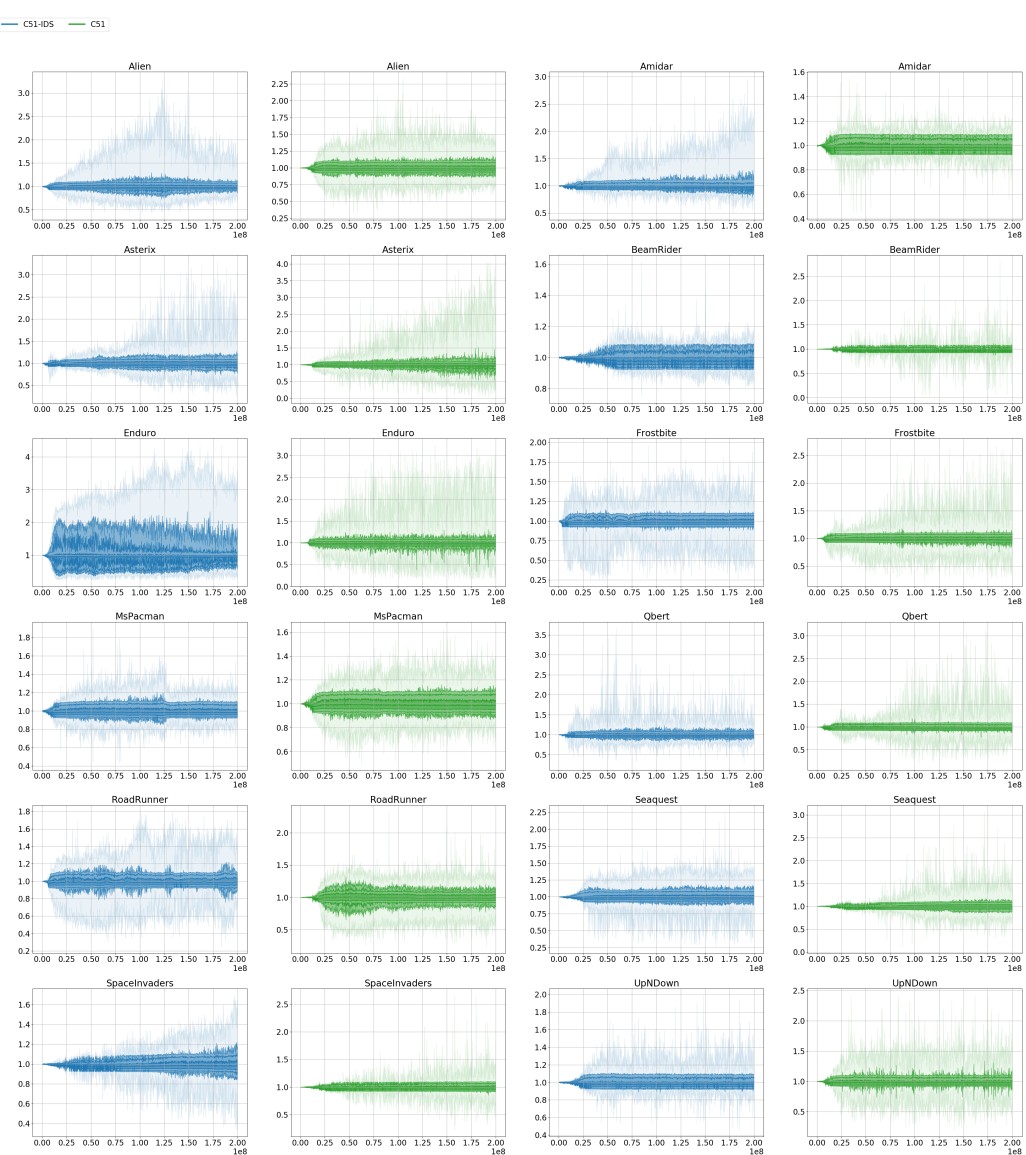

Figure 3: The return distributions learned by C51-IDS and C51. Plots obtained by sampling a random batch of 32 states from the replay buffer every 50000 steps and computing the estimates for $\rho^2(\mathbf{s}, \mathbf{a})$ based on eq. (9). A histogram over the resulting values is then computed and displayed as a distribution (by interpolation). From top to bottom, the lines on each plot correspond to standard deviation boundaries of a normal distribution $[\max, \mu + 1.5\sigma, \mu + \sigma, \mu + 0.5\sigma, \mu, \mu - 0.5\sigma, \mu - \sigma, \mu - 1.5\sigma, \min]$. The $x$-axis indicates number of training frames.

Table 3:  Raw evaluation scores. Episodes start with up to 30 no-op actions. Reference values from Wang et al. (2016) and Osband et al. (2016a).  DQN-IDS averaged over 3 seeds.  Bootstrap DQN scores for Berzerk, Phoenix, Pitfall!, Skiing, Solaris and Yars' Revenge obtained from our custom implementation.

| | DQN | DDQN | Duel. | Bootstrap | Prior.Duel. | DQN-IDS |
|---|---|---|---|---|---|---|
| Alien | 1,620.0 | 3,747.7 | 4,461.4 | 2,436.6 | 3,941.0 | **9,780.1** |
| Amidar | 978.0 | 1,793.3 | 2,354.5 | 1,272.5 | 2,296.8 | **2,457.0** |
| Assault | 4,280.4 | 5,393.2 | 4,621.0 | 8,047.1 | **11,477.0** | 9,446.7 |
| Asterix | 4,359.0 | 17,356.5 | 28,188.0 | 19,713.2 | **375,080.0** | 50,167.3 |
| Asteroids | 1,364.5 | 734.7 | **2,837.7** | 1,032.0 | 1,192.7 | 1,959.7 |
| Atlantis | 279,987.0 | 106,056.0 | 382,572.0 | **994,500.0** | 395,762.0 | 993,212.5 |
| Bank Heist | 455.0 | 1,030.6 | **1,611.9** | 1,208.0 | 1,503.1 | 1,226.1 |
| Battle Zone | 29,900.0 | 31,700.0 | 37,150.0 | 38,666.7 | 35,520.0 | **67,394.2** |
| Beam Rider | 8,627.5 | 13,772.8 | 12,164.0 | 23,429.8 | 30,276.5 | **30,426.6** |
| Berzerk | 585.6 | 1,225.4 | 1,472.6 | 1,077.9 | 3,409.0 | **4,816.2** |
| Bowling | 50.4 | **68.1** | 65.5 | 60.2 | 46.7 | 50.7 |
| Boxing | 88.0 | 91.6 | 99.4 | 93.2 | 98.9 | **99.9** |
| Breakout | 385.5 | 418.5 | 345.3 | **855.0** | 366.0 | 600.1 |
| Centipede | 4,657.7 | 5,409.4 | 7,561.4 | 4,553.5 | **7,687.5** | 5,860.2 |
| Chopper Command | 6,126.0 | 5,809.0 | 11,215.0 | 4,100.0 | 13,185.0 | **13,385.4** |
| Crazy Climber | 110,763.0 | 117,282.0 | 143,570.0 | 137,925.9 | 162,224.0 | **194,935.7** |
| Demon Attack | 12,149.4 | 58,044.2 | 60,813.3 | 82,610.0 | 72,878.6 | **130,687.2** |
| Double Dunk | -6.6 | -5.5 | 0.1 | **3.0** | -12.5 | 1.2 |
| Enduro | 729.0 | 1,211.8 | 2,258.2 | 1,591.0 | 2,306.4 | **2,358.2** |
| Fishing Derby | -4.9 | 15.5 | **46.4** | 26.0 | 41.3 | 45.2 |
| Freeway | 30.8 | 33.3 | 0.0 | 33.9 | 33.0 | **34.0** |
| Frostbite | 797.4 | 1,683.3 | 4,672.8 | 2,181.4 | **7,413.0** | 5,884.3 |
| Gopher | 8,777.4 | 14,840.8 | 15,718.4 | 17,438.4 | **104,368.2** | 47,826.2 |
| Gravitar | 473.0 | 412.0 | 588.0 | 286.1 | 238.0 | **771.0** |
| H.E.R.O. | 20,437.8 | 20,818.2 | **23,037.7** | 21,021.3 | 21,036.5 | 15,165.4 |
| Ice Hockey | -1.9 | -2.7 | 0.5 | -1.3 | -0.4 | **1.7** |
| James Bond | 768.5 | 1,358.0 | 1,312.5 | 1,663.5 | 812.0 | **1,782.2** |
| Kangaroo | 7,259.0 | 12,992.0 | 14,854.0 | 14,862.5 | 1,792.0 | **15,364.5** |
| Krull | 8,422.3 | 7,920.5 | **11,451.9** | 8,627.9 | 10,374.4 | 10,587.3 |
| Kung-Fu Master | 26,059.0 | 29,710.0 | 34,294.0 | 36,733.3 | **48,375.0** | 38,113.5 |
| Montezuma's Revenge | 0.0 | 0.0 | 0.0 | **100.0** | 0.0 | 0.0 |
| Ms. Pac-Man | 3,085.6 | 2,711.4 | 6,283.5 | 2,983.3 | 3,327.3 | **7,273.7** |
| Name This Game | 8,207.8 | 10,616.0 | 11,971.1 | 11,501.1 | 15,572.5 | **15,576.7** |
| Phoenix | 8,485.2 | 12,252.5 | 23,092.2 | 14,964.0 | 70,324.3 | **176,493.2** |
| Pitfall! | -286.1 | -29.9 | **0.0** | **0.0** | **0.0** | **0.0** |
| Pong | 19.5 | 20.9 | **21.0** | 20.9 | 20.9 | **21.0** |
| Private Eye | 146.7 | 129.7 | 103.0 | **1,812.5** | 206.0 | 201.1 |
| Q*Bert | 13,117.3 | 15,088.5 | 19,220.3 | 15,092.7 | 18,760.3 | **26,098.5** |
| River Raid | 7,377.6 | 14,884.5 | 21,162.6 | 12,845.0 | 20,607.6 | **27,648.3** |
| Road Runner | 39,544.0 | 44,127.0 | **69,524.0** | 51,500.0 | 62,151.0 | 59,546.2 |
| Robotank | 63.9 | 65.1 | 65.3 | 66.6 | 27.5 | **68.6** |
| Seaquest | 5,860.6 | 16,452.7 | 50,254.2 | 9,083.1 | 931.6 | **58,909.8** |
| Skiing | -13,062.3 | -9,021.8 | -8,857.4 | -9,413.2 | -19,949.9 | **-7,415.3** |
| Solaris | 3,482.8 | 3,067.8 | 2,250.8 | **5,443.3** | 133.4 | 2,086.8 |
| Space Invaders | 1,692.3 | 2,525.5 | 6,427.3 | 2,893.0 | 15,311.5 | **35,422.1** |
| Star Gunner | 54,282.0 | 60,142.0 | 89,238.0 | 55,725.0 | **125,117.0** | 84,241.0 |
| Tennis | 12.2 | -22.8 | 5.1 | 0.0 | 0.0 | **23.6** |
| Time Pilot | 4,870.0 | 8,339.0 | 11,666.0 | 9,079.4 | 7,553.0 | **13,464.8** |
| Tutankham | 68.1 | 218.4 | 211.4 | 214.8 | 245.9 | **265.5** |
| Up and Down | 9,989.9 | 22,972.2 | 44,939.6 | 26,231.0 | 33,879.1 | **85,903.5** |
| Venture | 163.0 | 98.0 | **497.0** | 212.5 | 48.0 | 389.1 |
| Video Pinball | 196,760.4 | 309,941.9 | 98,209.5 | **811,610.0** | 479,197.0 | 696,914.0 |
| Wizard Of Wor | 2,704.0 | 7,492.0 | 7,855.0 | 6,804.7 | 12,352.0 | **19,267.9** |
| Yars' Revenge | 18,098.9 | 11,712.6 | 49,622.1 | 17,782.3 | **69,618.1** | 25,279.5 |
| Zaxxon | 5,363.0 | 10,163.0 | 12,944.0 | 11,491.7 | 13,886.0 | **16,789.2** |

Table 4: Raw evaluation scores. Episodes start with up to 30 no-op actions. Reference values (available for a single seed) for C51, QR-DQN and IQN taken from Dabney et al. (2018b) and Dabney et al. (2018a). C51-IDS averaged over 3 seeds.

| | Random | Human | C51 | QR-DQN | IQN | C51-IDS |
|---|---|---|---|---|---|---|
| Alien | 227.8 | 7,127.7 | 3,166.0 | 4,871.0 | 7,022.0 | **11,473.6** |
| Amidar | 5.8 | 1,719.5 | 1,735.0 | 1,641.0 | **2,946.0** | 1,757.6 |
| Assault | 222.4 | 742.0 | 7,203.0 | 22,012.0 | **29,091.0** | 21,829.1 |
| Asterix | 210.0 | 8,503.3 | 406,211.0 | 261,025.0 | 342,016.0 | **536,273.0** |
| Asteroids | 719.1 | **47,388.7** | 1,516.0 | 4,226.0 | 2,898.0 | 2,549.1 |
| Atlantis | 12,850.0 | 29,028.1 | 841,075.0 | 971,850.0 | 978,200.0 | **1,032,150.0** |
| Bank Heist | 14.2 | 753.1 | 976.0 | 1,249.0 | **1,416.0** | 1,338.3 |
| Battle Zone | 2,360.0 | 37,187.5 | 28,742.0 | 39,268.0 | 42,244.0 | **66,724.0** |
| Beam Rider | 363.9 | 16,926.5 | 14,074.0 | 34,821.0 | **42,776.0** | 42,196.7 |
| Berzerk | 123.7 | 2,630.4 | 1,645.0 | 3,117.0 | 1,053.0 | **23,227.3** |
| Bowling | 23.1 | **160.7** | 81.8 | 77.2 | 86.5 | 57.0 |
| Boxing | 0.1 | 12.1 | 97.8 | **99.9** | 99.8 | **99.9** |
| Breakout | 1.7 | 30.5 | **748.0** | 742.0 | 734.0 | 575.5 |
| Centipede | 2,090.9 | 12,017.0 | 9,646.0 | **12,447.0** | 11,561.0 | 9,840.5 |
| Chopper Command | 811.0 | 7,387.8 | 15,600.0 | 14,667.0 | **16,836.0** | 12,309.5 |
| Crazy Climber | 10,780.5 | 35,829.4 | 179,877.0 | 161,196.0 | 179,082.0 | **205,629.6** |
| Demon Attack | 152.1 | 1,971.0 | **130,955.0** | 121,551.0 | 128,580.0 | 129,667.5 |
| Double Dunk | -18.6 | -16.4 | 2.5 | **21.9** | 5.6 | 1.2 |
| Enduro | 0.0 | 860.5 | **3,454.0** | 2,355.0 | 2,359.0 | 2,370.1 |
| Fishing Derby | -91.7 | -38.7 | 8.9 | 39.0 | 33.8 | **49.8** |
| Freeway | 0.0 | 29.6 | 33.9 | **34.0** | **34.0** | **34.0** |
| Frostbite | 65.2 | 4,334.7 | 3,965.0 | 4,384.0 | 4,324.0 | **10,924.1** |
| Gopher | 257.6 | 2,412.5 | 33,641.0 | 113,585.0 | 118,365.0 | **123,337.5** |
| Gravitar | 173.0 | **3,351.4** | 440.0 | 995.0 | 911.0 | 885.5 |
| H.E.R.O. | 1,027.0 | 30,826.4 | **38,874.0** | 21,395.0 | 28,386.0 | 17,545.3 |
| Ice Hockey | -11.2 | **0.9** | -3.5 | -1.7 | 0.2 | -0.5 |
| James Bond | 29.0 | 302.8 | 1,909.0 | 4,703.0 | **35,108.0** | 9,687.0 |
| Kangaroo | 52.0 | 3,035.0 | 12,853.0 | 15,356.0 | 15,487.0 | **16,143.5** |
| Krull | 1,598.0 | 2,665.5 | 9,735.0 | **11,447.0** | 10,707.0 | 10,454.5 |
| Kung-Fu Master | 258.5 | 22,736.3 | 48,192.0 | **76,642.0** | 73,512.0 | 59,710.7 |
| Montezuma's Revenge | 0.0 | **4,753.3** | 0.0 | 0.0 | 0.0 | 0.0 |
| Ms. Pac-Man | 307.3 | **6,951.6** | 3,415.0 | 5,821.0 | 6,349.0 | 6,616.2 |
| Name This Game | 2,292.3 | 8,049.0 | 12,542.0 | 21,890.0 | **22,682.0** | 15,248.1 |
| Phoenix | 761.4 | 7,242.6 | 17,490.0 | 16,585.0 | 56,599.0 | **89,050.8** |
| Pitfall! | -229.4 | **6,463.7** | 0.0 | 0.0 | 0.0 | 0.0 |
| Pong | -20.7 | 14.6 | 20.9 | **21.0** | **21.0** | **21.0** |
| Private Eye | 24.9 | **69,571.3** | 15,095.0 | 350.0 | 200.0 | 150.0 |
| Q*Bert | 163.9 | 13,455.0 | 23,784.0 | **572,510.0** | 25,750.0 | 27,844.0 |
| River Raid | 1,338.5 | 17,118.0 | 17,322.0 | 17,571.0 | 17,765.0 | **30,637.1** |
| Road Runner | 11.5 | 7,845.0 | 55,839.0 | **64,262.0** | 57,900.0 | 61,550.3 |
| Robotank | 2.2 | 11.9 | 52.3 | 59.4 | 62.5 | **69.8** |
| Seaquest | 68.4 | 42,054.7 | **266,434.0** | 8,268.0 | 30,140.0 | 86,989.3 |
| Skiing | -17,098.1 | **-4,336.9** | -13,901.0 | -9,324.0 | -9,289.0 | -7,785.4 |
| Solaris | 1,236.3 | **12,326.7** | 8,342.0 | 6,740.0 | 8,007.0 | 3,571.3 |
| Space Invaders | 148.0 | 1,668.7 | 5,747.0 | 20,972.0 | 28,888.0 | **46,244.2** |
| Star Gunner | 664.0 | 10,250.0 | 49,095.0 | 77,495.0 | 74,677.0 | **137,453.6** |
| Tennis | -23.8 | -8.3 | 23.1 | **23.6** | **23.6** | 23.5 |
| Time Pilot | 3,568.0 | 5,229.2 | 8,329.0 | 10,345.0 | 12,236.0 | **14,351.4** |
| Tutankham | 11.4 | 167.6 | 280.0 | **297.0** | 293.0 | 200.2 |
| Up and Down | 533.4 | 11,693.2 | 15,612.0 | 71,260.0 | 88,148.0 | **109,045.9** |
| Venture | 0.0 | 1,187.5 | **1,520.0** | 43.9 | 1,318.0 | 495.6 |
| Video Pinball | 16,256.9 | 17,667.9 | **949,604.0** | 705,662.0 | 698,045.0 | 756,111.1 |
| Wizard Of Wor | 563.5 | 4,756.5 | 9,300.0 | 25,061.0 | **31,190.0** | 18,817.4 |
| Yars' Revenge | 3,092.9 | 54,576.9 | 35,050.0 | 26,447.0 | 28,379.0 | **64,822.9** |
| Zaxxon | 32.5 | 9,173.3 | 10,513.0 | 13,112.0 | **21,772.0** | 18,295.4 |

