# OpenReview forum: "Information-Directed Exploration for Deep Reinforcement Learning"
_ICLR.cc/2019/Conference_

### Official Review · AnonReviewer1 · 2018-10-31
**The main input of this paper is to combine Information Direct Sampling and Distributional Reinforcement Learning for handling heteroscedasticity of noise in Reinforcement Learning.**

**Rating:** 7
**Confidence:** 4

**Review:**

This paper investigates sophistical exploration approaches for reinforcement learning. Motivated by the fact that most of bandit algorithms do not handle heteroscedasticity of noise, the authors built on Information Direct Sampling and on Distributional Reinforcement Learning to propose a new exploration algorithm family. Two versions of the exploration strategy are evaluated against the state-of-the-art on Atari games: DQN-IDS for homoscedatic noise and C51-IDS for heteroscedastic noise.

The paper is well-written. The background section provides the clues to understand the approach. In IDS, the selected action is the one that minimizes the ratio between a squared conservative estimate of the regret and the information gain. Following (Ktischner and Krause 2018), the authors propose to use \log(1+\sigma^2_t(a)/\rho^2(a)) as the information gain function, which corresponds to a Gaussian prior, where \sigma^2_t is the variance of the parametric estimate of E[R(a)] and \rho^2(a) is the variance of R(a). \sigma^2_t is evaluated by bootstrap (Boostrapped DQN). Where the paper becomes very interesting is that recent works on distributional RL allow to evaluate \rho^2(a). This is the main input of this paper: combining two recent approaches for handling heteroscedasticity of noise in Reinforcement Learning.

Major concern:
While the approach is appealing for handling heteroscedastic noise, the use of a normalized variance (eq 9) and a lower bound of variance (page 7) reveal that the approach needs some tuning which is not theoretically founded.
This is problematic since in reinforcement learning, the environment is usually assumed to be unknown. What are the results when the lower bound of the variance is not used? When the variance of Z(a) is low, the variance of the parametric estimate should be low also. It is not the case?


Minor concerns:

The color codes of Figure 1 are unclear. The color of curves in subfigures (b) (c) (d) corresponds to the color code of IDS.

The way in which \rho^2(s,a) is computed in algorithm 1 is not precisely described. In particular page 6, the equation \rho^2(s,a)=Var(Z_k(s,a)) raises some questions: Is \rho evaluated for a particular bootstrap k or is \rho is averaged over the K bootstraps ?
_____________________________________________________________________________________________________________________________________________

I read the answers of authors. I increased my rating.

---

> ### Author Response · Authors · 2018-11-20
> **Author's response**
>
> Thank you for the review and the comments.
>
> Please note that in the meantime, we were able to run experiments on 55 of the Atari games. The new results support our initial findings and are included in updated version of the paper.
>
> It is correct that in the bandit setting, the information-gain function with the unnormalized noise function (rho(s,a)) leads to the correct scaling of the regret-information ratio, such that the regret of IDS can be bounded. However it is not clear that this is necessarily the right choice when used in combination with deep reinforcement learning. In fact, the scaling of the reward differs significantly from game to game, which leads to different noise levels and values for the information gain function \log(1+\sigma^2_t(s, a)/\rho^2(s, a)). We found that the normalized noise estimation yields better results and allows the agent to account for numerical differences across environments while favoring the same amount of risk across different games. Importantly, it preserves the signal needed for noise-sensitive exploration
> and does not introduce a new tuning parameter. It also does not necessarily loosen the connection to IDS, which explicitly allows to design policies by using different information-gain functions.
>
> The lower bound on the return variance was introduced only for numerical reasons. Further, it  prevents the agent from overcommitting to low-variance actions. Even in the bandit case, the strategy degenerates as the noise variance of a single action goes to zero (because that way, the information gain of any action can be made arbitrarily large). Also, since the return variance is normalized, the values of return variance of different actions are relatively close to 1. Hence, a lower bound of 0.25 would not introduce significant difference. We note that we did not tune at all this value and selected it heuristically. We also conducted experiments without the lower bound on rho. While the per-game scores may slightly differ, the overall change in mean human-normalized score was only 23%. This is added to the revised version of the paper.
>
> To clarify the way in which \rho(s, a)^2 is computed in Algorithm 1: The bootstrap heads are used only to compute the predictive parametric uncertainty \sigma(s,a)^2. The return uncertainty \rho(s,a)^2 is computed based only on the output Z(s,a) of the distributional head. We have added the exact formula for Var(Z(s,a)) at the end of page 6 in the paper.
>
> Can you also please clarify your note about the color codes in Figure 1?

---

### Official Review · AnonReviewer3 · 2018-11-02

**Rating:** 7
**Confidence:** 3

**Review:**

The authors propose a way of extending Information-Directed Sampling (IDS) to reinforcement learning. The proposed approach uses Bootstrapped DQN to estimate parametric uncertainty in Q-values, and distributional RL to estimate intrinsic uncertainty in the return. The two types of uncertainty are combined to obtain a simple exploration strategy based on IDS. The approach outperforms a number of strong baselines on a subset of 12 Atari 2600 games.

Clarity - I found the paper to be very well-written and easy to follow. Both the background material and the experimental setup were explained very clearly. The main ideas were also motivated quite well. It would have been nice to include a bit more discussion of why IDS is a good strategy, i.e. what are the theoretical guarantees in the bandit case? Section 3.2 could also provide a more intuitive argument.

Novelty - The paper essentially combines the IDS formulation of Kirschner & Krause, Bootstrapped DQN of Osband et al., and the C51 distributional RL method of Bellemare et al. Most of the novelty is in how to combine these ideas effectively in the deep RL setting, which I found sufficient.

Significance - Improving over existing exploration strategies for deep RL would be a significant achievement. While the results are impressive, I have a few concerns regarding some of the claims.

The subset of games used to evaluate the proposed approach seems to be biased towards games where there is either a dense reward or exploration is known to be easy. Almost every deep RL paper on exploration includes results for at least some of the hard exploration games (see “Unifying Count-Based Exploration and Intrinsic Motivation”). Why were these games excluded from the evaluation? The results would be much stronger if results on all 57 games were included.

The main difference between DQN-IDS and C51-IDS is that C51-IDS will tend to favor actions with lower return uncertainty. Doesn’t this mean that the improved performance of C51-IDS is due to an improved ability to exploit rather than explore? If this is indeed the case, then I would expect more evidence that this doesn't come at a cost of reduced performance on tasks where exploration is difficult.

Finally, the comparison between Bootstrapped DQN and DQN-IDS conflates the exploration strategies (IDS vs Thompson sampling) with the choice of optimizer (Adam vs RMSProp), so the claim that simply changing the exploration strategy to IDS leads to a major improvement is not valid. It would be interesting to see results for Bootstrapped DQN using the authors’ implementation and choice of optimizer to fully separate the effect of the exploration strategy.

Overall quality - This is an interesting paper with some promising results. I’m not convinced that the proposed method leads to better exploration, but I think it still makes a valuable contribution to the work on balancing exploration and exploitation in RL.

-------

The rebuttal and revisions addressed some of my concerns so I am increasing my score to 7

---

> ### Author Response · Authors · 2018-11-20
> **Author's response**
>
> Thank you for the review and the comments.
>
> We first like to report, that in the meantime we were able to run our experiments on 55 Atari games simulated via the OpenAI gym interface. The result table is updated in the revised version of our paper and supports our initial findings: The homoscedastic DQN-IDS achieves a score of 757;187 (%mean; %median), and the heteroscedastic C51-IDS achieves 1058;253 which is competitive with IQN (1048; 218).
>
> Regarding the concern that the gain of C51-IDS is due to more exploitative actions: It is true that the main difference between DQN-IDS and C51-IDS is that C51-IDS tends to favor actions with lower return uncertainty (risk). However, the improved performance is unlikely to be due to more extensive exploitation. First of all, the results in Table 1, 3 and 4 are based on evaluation scores. These evaluation scores are obtained by running the agents with an evaluation policy which is computed in the same way for both DQN-IDS and C51-IDS and acts greedily w.r.t. the mean of all bootstrapped heads (Eq. 10). If C51-IDS was only focusing at exploitation during training (i.e. the data-collection process, while the IDS policy is being run), it would not be able to explore sufficiently and would likely converge to a suboptimal policy. Hence we would observe worse evaluation scores compared to DQN-IDS, which is not the case demonstrated by the overall results. Furthermore, even though actions with lower return uncertainty have higher information gain (as computed by C51-IDS), this does not necessarily lead to exploitation, as the choice additionally depends on the amount of parametric uncertainty as well as the ratio between regret and information gain (see also the Gaussian process example in Fig. 1). Additionally, it is not necessarily true that an action with a lower return uncertainty would be the greedy one.
>
> In terms of the comparison between Bootstrapped DQN and DQN-IDS, we previously ran some experiments on Bootstrapped DQN using the Adam optimizer and observed very little difference compared to RMSProp. We agree that a fair comparison would require running Bootstrapped DQN with the Adam Optimizer. We have corrected our claim in the paper. However, since this is not the focus of our paper and given the available computational resources, we will be unable to include Bootstrapped DQN results with the Adam optimizer over all 57 Atari games. We will also release the code after the final decision, which includes our implementation of Bootstrapped DQN.

---

### Official Review · AnonReviewer2 · 2018-11-04
**Good idea, well described, could use more experimental results**

**Rating:** 7
**Confidence:** 4

**Review:**

Combining the parametric uncertainty of bootstrapped DQN with the return uncertainty of C51, the authors propose a deep RL algorithm that can explore in the presence of heteroscedasticity. The motivation is quite well written, going through IDS and the approximations in a way that didn't presume prior familiarity.

The core idea seems quite sound, but the fact that the distributional loss can't be propagated through the full network is troubling. The authors' choice of bootstrapped DQN feels arbitrary, as a different source of parametric uncertainty might be more compatible (e.g. noisy nets), and this possibility isn't discussed.

The computational limitations are understandable, but the authors should be more transparent about how the subset of games were selected. A toy example would have actually added quite a bit, as it would nice to see that the extent to which this algorithm helps is proportional to the heteroscedasticity in the environment. The advantage of DQN-IDS over bootstrapped suggests that something other than just the sensitivity to return variance is causing these improvements.

Ideally, results with and without the heuristically chosen lower bound (rho) would be presented, as its unclear how much this is needed and its presence loosens the connection to IDS.

This is a small point, but the treatment of intrinsic motivation (i.e. changing the reward function) for exploration seems overly harsh. Most of these methods are amenable to experience replay, which would propagate the exploration signals and allow for "deep" exploration. The fact that they often change the optimal policy should be enough motivation to not discuss them further.

EDIT: I think dealing with the lower bound and including plots for all 55 games pushed this over the edge. It would've been nice if there non-zero scores on Montezuma's Revenge, but I know that is a high bar for a general purpose exploration method. In general I think this approach shows great promise going forward score 6-->7

---

> ### Author Response · Authors · 2018-11-20
> **Author's response**
>
> Thank you for the review and the suggestions.
>
> We first like to report, that in the meantime we were able to run our experiments on 55 Atari games simulated via the OpenAI gym interface. The result table is updated in the revised version of our paper and supports our initial findings: The homoscedastic DQN-IDS achieves a score of 757;187 (%mean; %median), and the heteroscedastic C51-IDS achieves 1058;253 which is competitive with IQN (1048; 218).
>
> To clarify the concern raised on propagating the distributional loss: We emphasize that we chose not to propagate the distributional loss into the full C51-IDS network and use the C51 distribution only for control. This allows us to isolate the effect of noise-sensitive exploration and gives a fair comparison between DQN-IDS and C51-IDS. This is not a limitation of our approach and we would expect an additional performance gain by propagating distributional gradients computed on a distributional loss like C51 or QR-DQN. This remark has been added to the paper.
>
> We also conducted experiments without the lower bound on rho. While the per-game scores may slightly differ, the overall change in mean human-normalized score was only 23%. This as well is mentioned in the revised version of the paper.
>
> In terms of the choice of parametric uncertainty estimator, we selected Bootstrapped DQN since it allows computing the predictive distribution variance, without the need for any sampling. We also briefly experimented with Neural Bayesian Linear Regression (Snoek et al, 2015), but we found Bootstrapped DQN to yield better results. However, as discussed in the related work section, we acknowledge there are other ways of estimating parametric uncertainty, such as noisy nets, Monte Carlo methods, Bayesian Dropout, etc.
>
> The comparison to intrinsic motivation is re-phrased in the updated version of the paper.

---

### Public Comment · ~Ankesh_Anand1 · 2018-10-02
**Choice of Atari environments**

I like the idea of this paper of extending Information-Directed Sampling to large state spaces. I also appreciate the computational constraints, and why the authors decided to test on only 12 Atari environments, but I was a bit perplexed by the choice of environments. Shouldn't the games that are known particularly to be hard to explore, such as Montezuma's Revenge, Pitfall and PrivateEye been evaluated? The games that the paper tested on not actually hard exploration problems (except Frostbite arguably).

---

> ### Author Response · Authors · 2018-10-18
> **Response to comment**
>
> Thank you for the comment. The currently reported range of games was chosen in the following way. We first selected 3 games on which convergence was relatively quick (BeamRider, RoadRunner, Enduro) so that we can more easily tune our algorithm. The rest of the games were chosen as a combination of games on which Bootstrapped DQN and C51 achieve improvement over the baseline. In particular, we wanted to evaluate the homoscedastic version of our algorithm (DQN-IDS) against the best scores that Bootstrapped DQN achieves. Additionally, high C51 scores indicate that C51 achieves good estimate of the return distribution and we wanted to test whether our algorithm (C51-IDS) can benefit from this and improve over C51. Note that the selection also includes games on which C51 achieves poor results.
>
> We are currently in the process of evaluating our method on more games, and we expect to get further results until the rebuttal period. The scores will be included in the revised version of the paper.

---

### Meta-Review · Area_Chair1 · 2018-12-13
**Well written paper with an novel approach for exploration**

**Confidence:** 4
**Recommendation:** Accept (Poster)

**Metareview:**

The paper introduces a method for using information directed sampling, by taking advantage of recent advances in computing parametric uncertainty and variance estimates for returns. These estimates are used to estimate the information gain, based on a formula from (Kirschner & Krause, 2018) for the bandit setting. This paper takes these ideas and puts them together in a reasonably easy-to-use and understandable way for the reinforcement learning setting, which is both nontrivial and useful. The work then demonstrates some successes in Atari. Though it is of course laudable that the paper runs on 57 Atari games, it would make the paper even stronger if a simpler setting (some toy domain) was investigated to more systematically understand this approach and some choices in the approach.